# Returning Individual Tap Water Testing Results to Research Study Participants after a Wildfire Disaster

**DOI:** 10.3390/ijerph19020907

**Published:** 2022-01-14

**Authors:** Julie Von Behren, Michelle Wong, Daniela Morales, Peggy Reynolds, Paul B. English, Gina Solomon

**Affiliations:** 1Department of Epidemiology and Biostatistics, University of California San Francisco, San Francisco, CA 94158, USA; peggy.reynolds@ucsf.edu; 2Tracking California, Public Health Institute, Oakland, CA 94607, USA; michelle.wong@trackingcalifornia.org (M.W.); danielamorales@berkeley.edu (D.M.); paul.english@trackingcalifornia.org (P.B.E.); 3Department of Medicine, University of California San Francisco, San Francisco, CA 94158, USA; gina.solomon@phi.org; 4Public Health Institute, Oakland, CA 94607, USA

**Keywords:** environmental health, results communication, emergency response, disaster research, wildfire, drinking water, California

## Abstract

After the devastating wildfire that destroyed most of the town of Paradise, California in 2018, volatile organic compounds were found in water distribution pipes. Approximately 11 months after the fire, we collected tap water samples from 136 homes that were still standing and tested for over 100 chemicals. Each participant received a customized report showing the laboratory findings from their sample. Our goal was to communicate individual water results and chemical information rapidly in a way that was understandable, scientifically accurate, and useful to participants. On the basis of this process, we developed a framework to illustrate considerations and priorities that draw from best practices of previous environmental results return research and crisis communication, while also addressing challenges specific to the disaster context. We also conducted a follow-up survey on participants’ perceptions of the results return process. In general, participants found the results return communications to be understandable, and they felt less worried about their drinking water quality after receiving the information. Over one-third of the participants reported taking some kind of action around their water usage habits after receiving their results. Communication with participants is a critical element of environmental disaster research, and it is important to have a strategy to communicate results that achieves the goals of timeliness, clarity, and scientific accuracy, ultimately empowering people toward actions that can reduce exposure.

## 1. Introduction

On 8 November 2018, the Camp Fire in California burned 18,804 homes and buildings and caused the deaths of 85 people. About 1700 homes remained standing in the town of Paradise and the surrounding area. Due to previously reported chemical contamination in a drinking water system after the 2017 Tubbs Fire in northern California, a state agency and local water utilities tested water samples for volatile organic compounds (VOCs) [1]. VOCs were found within the water distribution pipes, with the highest concentrations—up to about 1000 times the maximum contaminant level (MCL) for benzene—in service lines to homes. Public testing did not include private property. The water utility in Paradise issued a “do not drink/do not use” advisory about 1 month after the fire, just as some residents were returning to their homes that were still standing [2]. Multiple media outlets covered the discovery of drinking water contamination, with attention from local, state-wide, and even national news [3,4,5]. There was a high level of community concern, with local residents demanding information from the water utility as they tried to decide if it was safe to return home.

Recognizing the growing need for environmental health research in response to disasters, the National Institute of Environmental Health Sciences (NIEHS) created the Disaster Research Response (DR2) Program [6,7]. The agency also has a funding mechanism for time-sensitive environmental health research in response to disasters. The program was established because “the knowledge that is generated through well-designed, effectively executed research in anticipation of, in the midst of, and after an emergency is critical to our future capacity to better achieve the overarching goals of preparedness and response: preventing injury, illness, disability, and death, and supporting recovery” [8].

Our team, supported by the NIEHS time-sensitive research mechanism, conducted a rapid exposure research project in Paradise and surrounding areas less than 1 year after the fire. We surveyed residents and collected tap water samples inside standing homes in the affected area in order to characterize potential exposures and health risks to returning residents. We also evaluated spatial patterns and concentrations of over 100 volatile organic compounds, as well as conducted nontargeted testing in some homes for many hundreds of additional contaminants that might help explain contamination sources. The data collection, chemical and spatial analyses, and study results are described in detail elsewhere [9]. A key component of our research project was the return of individual household drinking water sampling results to study participants. Because we tested for so many chemicals in the water samples, including many without regulatory benchmarks, there was a large amount of technical information to convey to each study participant.

There is extensive prior research on best practices for communicating personal research results related to environmental contaminants [10,11,12,13,14,15,16,17,18,19]. Those studies, however, often included a time-consuming community engagement process to tailor and test messages and materials, and those studies were not conducted in the immediate aftermath of an environmental disaster. Communication during emergencies is another field for which best practices have been established. The National Incident Management System (NIMS) and Incident Command System (ICS) include strategies for crisis communication during emergencies [20]. These systems emphasize the establishment of a Joint Information Center (JIC) to communicate official, timely, accurate, easy-to-understand, and consistent information from responding agencies to the public [21], but these communications are not designed to encompass individualized research results. Several researchers have outlined the keys steps and challenges for integrating research activities into responses to public health emergencies [8,22]. These frameworks acknowledge the importance of sharing research findings with the impacted community and even with study participants, but an approach to the challenge of quickly developing and distributing individual results return materials has not been proposed.

This report describes the rapid development of results communication materials and return of individual household water sampling results in the aftermath of a massively disruptive environmental disaster. We surveyed participants’ perceptions of the materials and gauged the usefulness of the results to the participants. We then developed a proposed set of principles for individual results return in environmental disaster research, according to existing best practices and our field experience. As environmental disasters increase in frequency with climate change, it will be increasingly important to rapidly develop individual results return materials that are useful to people as they make critical decisions about personal and family safety in the aftermath of tragedy.

## 2. Materials and Methods

We collected tap water samples from approximately 10% of the homes still standing within the burn zone of the Camp Fire. Two water utilities served the affected area, and homes served by either utility were eligible. Homes on private wells were excluded. Our study was reviewed and approved by the Institutional Review Board of the Public Health Institute (IRB# I19-020).

Participants were recruited at a community meeting and through flyers posted at local businesses that had reopened, social media, and email outreach. We also received recruitment assistance from the North Valley Community Foundation, Paradise Irrigation District, California Office of Emergency Services, State Water Resources Control Board, and the University of California Davis Environmental Health Sciences Center. Stories in the local newspaper and on public radio about the study also encouraged residents to participate. Additional recruitment was done by word-of-mouth and door-to-door efforts.

Sampling teams visited 136 homes in October and November of 2019 to obtain written participant informed consent, administer a questionnaire, and collect samples for analysis. Study personnel also collected observational information about the home, degree of fire damage, and water pipes. The questionnaire included questions about household tap water use, knowledge of potential contaminant issues, participant observations about the water (e.g., color, smell, taste), degree of concern about the quality of the water, and health symptoms.

Our main goal for the results return process was to communicate individual water results and information about potential health implications and health-protective actions rapidly in a way that was understandable, scientifically accurate, and useful to the participant. We also wanted our materials to be consistent with information that participants received from local water agencies, when feasible and appropriate. We were sensitive to participants’ anxiety about their water after this traumatic event, and we wanted to return results as quickly as possible without impacting accuracy or clarity. We developed an accelerated strategy for results communication (Figure 1).

Our goal was to provide results packets to individual participants within 2 weeks of receipt of the results from the laboratory and communicate aggregate preliminary results to the community within 3 weeks. Results communication included a packet of written materials for each study participant with their personalized results, a three-page frequently asked questions (FAQ) document, and factsheets on commonly-detected contaminants. We then presented our general preliminary findings at a community meeting attended by at least 50 project participants and others at a local church. This meeting was live-streamed on the Paradise Irrigation District website, and the recording was available for viewing afterward. We also invited individual conversations with all study participants in whose water we detected any contaminant above a regulatory limit (maximum contaminant level), or any participant who wanted to discuss their results either by telephone or in person.

We reviewed public notices, fact sheets, web pages, and alerts that study participants may have received from state agencies and local water utilities. These materials included information on the ongoing public water system testing processes and results, impact of fire on water quality, and specific contaminants. We also reviewed background information on VOCs from state and federal government agencies, as well as coverage in local media. Attention to information participants may have received from other sources helped ensure consistent and relevant messaging, where appropriate.

The community meeting at which we recruited study participants was convened by the California Office of Emergency Services and included presentations on drinking water quality from one of the local water utilities and from the principal investigator of our study team (G.S.). Coordination with the emergency response agencies facilitated recruitment and access to information, but we were careful to emphasize the independence of our research project. The meeting was attended by over 100 local residents, and questions asked during the meeting or during one-on-one conversations after the meeting helped identify community concerns.

The draft results communication packet template was developed prior to receiving results from the laboratory, meaning that there was significant uncertainty about what results we would need to communicate. Regardless, a framework that included categorization of the results into four color-coded categories was developed as detailed below. Frequently asked questions (FAQs) were created on the basis of questions asked and concerns raised at the community meeting, during study recruitment, and during sampling visits. They also addressed similarities and differences between our sampling methods and those used by the water systems. Factsheets on anticipated common contaminants such as benzene and trihalomethanes were also developed in advance, while other contaminant-specific factsheets were developed once the laboratory analyses were completed. According to our discussions with emergency response and water agency staff and with community members, we knew that the participants were already aware of the potential water contamination and were likely to care about nuance and details of their water testing results due to the high degree of community concern and on-going media attention. Therefore, we decided to provide detailed text and information in the factsheets, FAQs, and individual results return reports. Draft materials were reviewed by the project team, health educators with Tracking California, the California Division of Drinking Water, and the IRB governing the research study. The draft materials were not pilot-tested with community members due to the rapid timeline for results communication.

As laboratory results were reported, each result was sorted according to which chemicals were detected and at what concentrations relative to benchmark values. We sent information packets to participants, starting with high-priority results (results with detections above or near an MCL). Individual results packets were sent to participants via email or postal mail, according to the preference recorded on their consent form. Using this process, we were generally able to achieve our goal of returning results within 2 weeks of receipt from the lab. Some residents had difficulty receiving their email results due to repeated power outages in and near the burn area, and postal deliveries were significantly delayed in the area. In several cases, residents did not receive their results promptly, and we needed to resend or even hand-deliver the results packets.

In addition to the questionnaire at the time of sample collection, we surveyed participants again 4 months after they received their household testing results. The survey was sent three times via an email link; participants without an email address were contacted by telephone for questionnaire administration. Study participants who we did not initially reach by telephone were recontacted three times to attempt to increase participation. This survey asked whether and how participants were using the tap water, whether there was any period of time when concern about tap water interfered with their daily life, and whether the study results affected their concern about the water. We also asked a series of questions about the results return process, inquiring in what ways they recalled interacting with the study team, as well as how understandable and useful the results were, and inviting them to rate their confidence in the results.

On the basis of this process and the feedback we received from study participants, we developed a framework to illustrate considerations and priorities that draw from the best practices of previous environmental results return research and crisis communication, while also addressing challenges specific to research in the disaster context.

## 3. Results

We obtained water samples and surveys from 136 households. The average age of the residents in the participating homes was 53 years, and 19% of the households had children under the age of 18 years living in the home. All participants from our study were non-Hispanic white. This older, white demographic reflects the census data, which indicate that the population in Paradise and the surrounding area is predominately non-Hispanic white (93%) and is older on average than the US population (49 years as compared to 38 years) [23,24]. Forty percent of the study participants did not choose email as their preferred method of communication, suggesting that this older, rural population was less technologically oriented than younger populations might be.

The average household income and education levels in the areas included in this investigation were somewhat lower than the California averages. The estimated median household income in the towns of Paradise and Magalia is 51,566 and 50,415 USD, respectively, but only 31,250 USD in the neighboring community of Concow, which was also included in our study area [24]. These median income levels are all lower than the California median household income of 75,000 USD. Overall, about 27% of the adult population in Butte County, the area of this study, had a 4 year college degree or higher, whereas, in the town of Magalia, 19% of adults had a college degree, somewhat lower than the California average of 34%.

At the time of water sample collection, 24% (32/136) of participants reported that they were using municipal water as their primary drinking source. In the town of Paradise, 90% of participants reported receiving a drinking water advisory, whereas, in surrounding communities, only 38% recalled receiving a water advisory. Furthermore, 47% of participants in Paradise responded that they were “very worried” about water safety, whereas 33% of participants in other nearby communities were “very worried”.

Each participant received a customized report, either via email or by US postal mail, showing the findings from their water sample (Figure 2). The results were color-coded to help participants understand and interpret the relative importance of the various chemical findings. For results that exceeded an MCL, we contacted residents by telephone to address individual questions about health concerns.

### 3.1. Factsheets about Specific Contaminants

In order to provide more context on chemicals that were found in the water, we prepared standardized factsheets (Appendix A) that were included with the individual results. Packets included factsheets on VOCs in drinking water and additional factsheets for specific contaminants (e.g., trihalomethanes, benzene, methylene chloride) based on individual lab results. We organized the factsheets by the following categories: (1) what is it; (2) how does it get in drinking water; (3) why is this a concern; (4) how to reduce exposure; (5) how is this regulated; (6) more information.

### 3.2. Evaluation of the Results Return and Participant Feedback

In March 2020, we conducted a follow-up survey of participants asking about their perceptions of the results return material. A total of 90 households completed the follow-up survey from the original 136 homes (66%). Study participants reported that they generally found their individual results materials to be “understandable” (mean = 8.3/10) and “useful” (mean = 8.7/10). Participants reported a high level of confidence in the results (mean = 8.7/10). Factors identified as inspiring confidence included the affiliations of the project team, the study funder, and interactions with the study team (Appendix A).

We asked a multiple-choice question about which communication methods were useful, allowing multiple responses. Over three-quarters of the participants (77%) responded that the written results packets were helpful. Other components identified as being helpful for understanding the results included the community meeting (21%) and direct interactions with the principal investigator (17%) or study staff (12%). Aspects of the study identified as useful included the water sample results (87%), background information about chemicals (54%), and other information in the individual results return packets (36%). One-quarter (26%) of the study participants reported that one of the aspects of the study most useful to them personally was the ability to participate in a research project (Appendix A).

Four months after receiving the study results, 56% of participants responded that they were much less worried about drinking water quality and 21% were slightly less worried; only two participants replied that they were more worried about their drinking water after participation in the study (Table 1). We asked participants about the general consistency of the information they received from multiple sources. Accordingly, 71% of respondents found the information generally consistent, whereas 11% responded that the information was inconsistent. Thirty-two participants reported taking action after receiving their water sampling results, including starting to use the tap water for showering (21 out of the 32 people who took action), starting to drink their tap water again (*N* = 12), and getting a water filter (*N* = 8). Four participants reported that they stopped using a water filter or water tank. Only one person reported that they started flushing water when it had been stagnant in the pipes, a recommendation made in our results return packets.

Due to the high degree of community concern over the quality of drinking water, we asked whether there had been a period of time when concern about the tap water interfered with daily life. Two-thirds of the participants replied yes; 80% of these reported a duration of more than 6 months, and 15% said that the concern about water interfered for 1–6 months. A total of 25 participants reported that their concern about the water still interfered with daily life more than 16 months after the fire.

Most of the participants who completed the follow-up survey (*N* = 77, 86%) received their water from the Paradise Irrigation District (PID), the largest provider in the area. Thirteen responding participants were served by the Del Oro Water Company (DOWC). The responses were generally similar between residents in the two districts, except that more DOWC participants than PID participants reported that the research study information was inconsistent with what they were hearing from their drinking water utility and state agencies (23% vs. 9%, respectively), which may have been due to the fact that the DOWC did not issue the same “do not drink/do not use” advisory that PID did. Another difference we noticed between the participants in the two districts was that only one (8%) DOWC resident reported taking any action after receiving their tap water sampling results, compared to 31 PID residents (40%).

The average age of the residents in the households that did not complete the follow-up survey was 48 years, which was similar to the responding household average age of 53 years. The participation rate was higher for residents served by the Paradise Irrigation District compared to the Del Oro Water Company (71% and 46%, respectively), likely due to the heightened level of concern about water quality among residents served by PID.

### 3.3. Framework for Returning Individual Results to Participants after a Disaster

According to our review of the existing literature and our experience conducting individual results return in the aftermath of a major disaster, we identified some priorities within environmental disaster research that overlap with, or differ from, individual results return in traditional research. We, therefore, developed a framework to illustrate considerations and priorities drawn from previous environmental health results return and crisis communication literature, while also addressing the additional challenges specific to conducting research in the disaster context (Figure 3).

Individual results return in the context of a disaster must incorporate, to the extent feasible, all best practices from environmental health results return, while also incorporating additional components. First, the participants have the right to know about the results from any samples collected from them, as well as the summary results from the community at large. These results must be conveyed with the appropriate degree of scientific certainty about the potential health implications. The results must be returned to participants in culturally and linguistically appropriate ways. Ideally, community input is sought throughout the process, including pretesting results return materials if time permits.

Key components of crisis communication strategies also need to be considered in disaster research results return. These factors include providing information in a timely and urgent manner. The messages should be simple, repeated, and factually accurate. The communications should provide honest information with empathy for those affected by the disaster. The researchers also need to be flexible and ready to respond to rapidly changing situations.

Research conducted during the response and recovery phases of a disaster requires unique consideration of factors that are not normally an issue in research studies, such as coordination with responders, not compromising or interfering with Incident Command, avoiding resource allocation conflicts (e.g., researchers should not compete with locals for food, water, shelter, healthcare, or security), infrastructure availability (e.g., electricity, postal delivery), research team safety, and the added vulnerability of affected community members. It is also difficult to involve the community in a formal engagement process because of the displacement caused by a disaster and the overwhelmed mental state of many of the participants. Disasters result in continually changing situations with a dynamic flow of information which requires special attention to information consistency when households are receiving multiple messages from different agencies, as well as researchers.

## 4. Discussion

The participants in our wildfire tap water research study found their individual results return communications to be understandable, useful, and confidence-inspiring. Receiving their household tap water sampling results made most participants feel less worried about their drinking water quality, despite the fact that our study did report finding some chemicals in most samples [9]. Over one-third of the participants reported taking some kind of action around water usage habits after receiving their tap water sampling results.

A large number of study participants reported that concerns about tap water still interfered with their daily life more than 1 year after the initial wildfire disaster. The high level of ongoing stress and anxiety in the community due to trauma from the wildfire was documented through another community survey conducted in 233 households in Paradise 6 months after the fire [23]. That survey found that a majority of households (54%) had at least one person in the home who experienced anxiety, stress, or depression due to concerns about tap water contamination. Other research reporting high rates of depression, post-traumatic stress disorder (PTSD), and anxiety in the local community after the wildfire underscores the challenge of returning results in a context of recent trauma [26].

The rapid nature of the development of our results return materials created several challenges. The short time frame for the research activities meant that there was no time to field-test the packet with a representative community, which limited our ability to fully implement community-engaged research principles. It was also difficult to connect with community organizations because of the devastation of homes, schools, churches, and businesses, as well as the displacement of much of the population. Prior to the fire, there were no nongovernmental or community-based organizations in the local area working on environmental health issues. Because we were testing water samples for over 100 different chemicals, it was not possible to predict the potential results in advance. Some results were unexpected and, therefore, more challenging to communicate with respect to health risks, scientific uncertainty, potential exposure sources, and ways to reduce exposure. The older, white demographic and generally low computer literacy of the residents in Paradise may limit the generalizability of our study to other communities. The participants in this study were all English speakers; thus, we did not have to develop translated materials, which would have required more time for development and review. The participants also had a high degree of literacy; as such, we could rely on written communications, a method that would not be as effective in communities with low literacy. We conducted our water sampling after the local water districts already completed their initial testing; therefore, we were able to assess the consistency between the previous messaging of water utilities and our communications, as well as explain any differences in our testing methodology. However, if we had been deployed into the field sooner, there would have been an added challenge to actively coordinate with the water agencies to ensure that results being released simultaneously were consistent and not confusing to community members.

Because people in the area were already highly concerned about water contamination, we erred on the side of providing more information and interpretation rather than less. This high level of detail in the results return communications may not be generalizable or appropriate for all communities. Depending on the community that is affected by a disaster, different issues may be faced that affect the rapid development of results return materials, such as different languages and cultures, varying literacy levels, levels of mistrust, and awareness of environmental health. In addition, the information needs of disaster survivors who have an immediate and vested interest in knowing their sampling results may be different from general research participants who may not have the same level of concern.

The return of individualized results to participants is an increasingly accepted and necessary part of conducting environmental health research. In the early 2000s, individual results return was considered controversial due to concerns about appropriately communicating uncertainty about the meaning of results and the potential to create additional stress and anxiety among study participants [27]. However, many investigators have found that participants in community-based environmental health research projects want to receive their results, find their results informative and useful, and sometimes take action on the basis of their results [12,13,14,15,17,18,27,28]. Today, results return processes are widely incorporated into environmental health research plans and were recommended in a 2018 report by the National Academies of Sciences, Engineering, and Medicine [29]. Other researchers have described principles and frameworks for guiding and improving results return communications in environmental health research. Lebow-Skelley identified five goals of effective results return: effective communication, community knowledge and concerns, uncertainty, empowering action, and institutional oversight [19]. Dunagan et al. (2013) highlighted 12 “tips” for reporting personal exposure results [11]. We were able to incorporate most of these goals and tips, with the exception of involving participants and community organizations throughout the study process and pretesting materials with study participants. This type of longitudinal community involvement is very difficult in the disaster setting due to community disruption and displacement. However, we were able to hold community meetings and give community presentations with summary results as recommended. We were able to follow the recommendations from Dunagan related to effective communication and conveying uncertainty. We explained the scientific limitations for our knowledge about the health implications, especially for the chemicals which had no regulatory benchmarks. We used colorful graphics in the personal reports. Another goal recommended for this type of results return relates to institutional review and the importance of continuing to educate IRBs about this type of community-based research. We engaged our IRB regarding our report back process and the importance of the participants’ right to know. We found, however, that additional factors also need to be considered in the results return process after a disaster. These include special attention to returning results quickly to participants so that they can use the information to make informed time-sensitive decisions, as well as being mindful of the effects of recent trauma and the importance of consistency with other messages. Personal connection between research staff and participants, as well as the use of multiple methods of communication, is particularly important in communities disrupted by a major disaster.

Disaster response research (DR2) emerged in the US within the last two decades in response to multiple disasters including the World Trade Center attacks (2001), Hurricanes Katrina (2005) and Sandy (2012), and the Deepwater Horizon BP Gulf oil spill (2010). NIEHS recognized the increasing frequency of disasters and the importance of developing methods to collect exposure data and conduct public health research in the aftermath of multiple types of disaster. Environmental exposure data have been a key focus of the NIEHS efforts, with the recognition that prospective cohort studies after disasters historically have suffered from exposure misclassification bias due to delays in collecting exposure information in the early phases of the disaster response [30]. The NIEHS DR2 program has drafted best practices for human subjects research [22], created a community of disaster researchers, and developed a repository of curated tools and resources for researchers [7]. Our experience shows that public health researchers should work in advance of disasters to develop and test multiple types of results return materials in multiple languages and with different literacy levels that can be used as templates for future events. These materials require a large investment of time and skill to create and test, which can be a limitation especially because most response teams do not have graphic designers or health education staff with experience in results communication.

## 5. Conclusions

Environmental health researchers will be increasingly called upon to conduct sampling and research that addresses community health concerns in response to environmental disasters. Communication with participants is a critical element of environmental disaster research, and it will be necessary to have a strategy to communicate results that achieves the goals of timeliness, clarity, and scientific accuracy. Existing strategies for results communication can be adapted to emergency situations, drawing on lessons from disaster risk communication, as well as presenting the need for contextual variations on these themes. One of the many research challenges of working in the aftermath of a disaster is the necessity of beginning work on results communication early in the research process, before results are available, and offering study participants multiple ways to interact with the research team for information. Successful communication in this setting can empower people toward action and may, therefore, help mitigate some of the trauma and assist with rebuilding a sense of community and personal agency.

## Figures and Tables

**Figure 1 ijerph-19-00907-f001:**
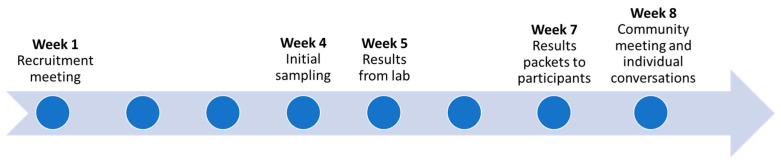
Timeline for results communication.

**Figure 2 ijerph-19-00907-f002:**
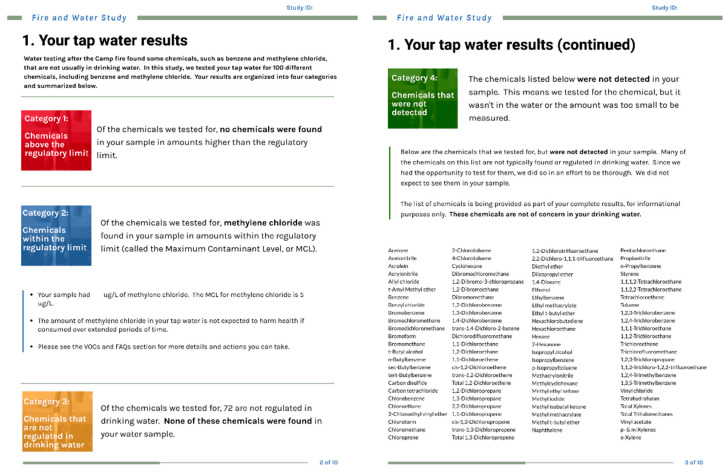
Example of a participant’s individualized results return report.

**Figure 3 ijerph-19-00907-f003:**
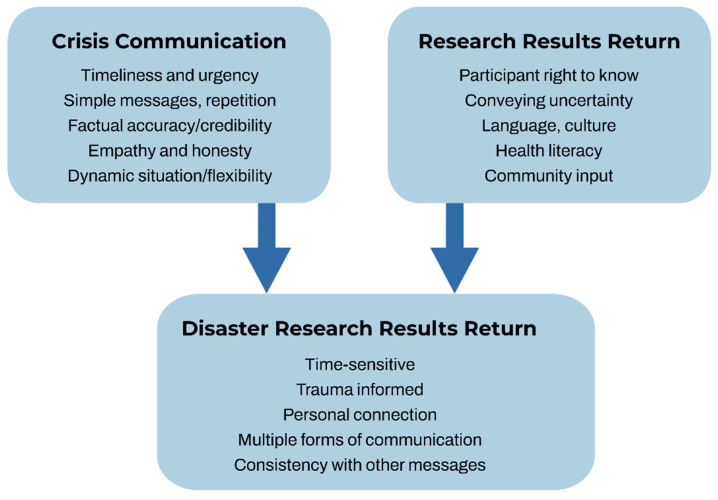
Framework for returning individual results to participants after a disaster adapted from Lebow-Skelley, 2020 [19] and CDC, 2002 [25].

**Table 1 ijerph-19-00907-t001:** Participant responses related to drinking water, Paradise California.

Survey Questions	Participants *N* = 90
Was the information from the research study consistent or inconsistent with what you were hearing from your drinking water utility and state agencies?	
Generally consistent	64 (71%)
Inconsistent	10 (11%)
Do not know	12 (13%)
Did not answer	4 (4%)
What difference has participation in the study made to your feelings about your drinking water?	
Much less worried about drinking water quality	50 (56%)
Slightly less worried about drinking water quality	19 (21%)
No change in my feelings around drinking water	15 (17%)
Slightly more worried about drinking water	1 (1%)
Much more worried about drinking water	1 (1%)
Unknown	4 (4%)
Did you take any action after receiving your tap water sampling results? Yes	32 (36%)
If yes, what did you do? (could choose more than one)	
Started using tap water in other ways (such as showering)	21 (66% among yes)
Started drinking the tap water	12 (38% among yes)
Got a water filter	8 (25% among yes)
Stopped using a filter or tank	4 (13% among yes)
Started flushing the water when it has been stagnant in the pipes	1 (3% among yes)
Still drink bottled water only	1 (3% among yes)
Got on list to have water company test water pipe from main to service line	1 (3% among yes)
Got a water tank	1 (3% among yes)
Between the fire and now, has there been a period of time when your concern about the tap water interfered with your daily life? Yes	59 (66%)
If yes, for about how long did your concern about the tap water interfere with your daily life?	
More than 6 months	47 (80% among yes)
More than 1 month to 6 months	9 (15% among yes)
More than 1 week to 1 month	2 (3% among yes)
1–7 days	1 (2% among yes)
If yes, does your concern about the water still interfere with your daily life?	25 (42% among yes)

## Data Availability

Data not available due to privacy rules and Institutional Review Board requirements.

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
