# Peer review of "Returning Individual Tap Water Testing Results to Research Study Participants after a Wildfire Disaster"

_ijerph, 2022, doi:10.3390/ijerph19020907_

Round 1

Reviewer 1 Report

The paper presents a framework for returning individual results to participants after an environmental disaster. The framework was developed based on the testing of household tap water after the wildfire disaster of 2018 in the Californian town of Paradise and the subsequent communication of results to individual study participants. The paper is well written, and the scientific method, discussion, developed framework and conclusion are sound and supported by the available data.

However, the following can be improved:

  1. Figure 3 can be improved; the text is difficult to read
  2. More detailed discussion/explanation of the Framework for Returning Individual Results to Participants after a Disaster. For example, what is meant by the point “language, culture”?

Author Response

We thank the reviewer for the helpful comments.

COMMENT: Figure 3 can be improved; the text is difficult to read   

RESPONSE: We have revised the figure to make it easier to read.

COMMENT: More detailed discussion/explanation of the Framework for Returning Individual Results to Participants after a Disaster. For example, what is meant by the point “language, culture”?

RESPONSE: This comment refers to part of Figure 3 that is not fully described in the text.  We added text to section 3.3 that describes all of the key points in our framework as shown in Figure 3.  The additional text is as follows: “First, the participants have the right to know about the results from any samples collected from them, as well as the summary results from the community at large. These results must be conveyed with the appropriate degree of scientific certainty about the potential health implications. The results must be returned to participants in culturally and linguistically appropriate ways. Ideally, community input is sought throughout the process, including pre-testing results return materials if time permits.   Key components of crisis communication strategies need to be considered in disaster research results return as well. These factors include providing information in a timely and urgent manner. The messages should be simple, repeated, and factually accurate. The communications should provide honest information with empathy for those affected by the disaster. The researchers also need to be flexible and ready to respond to rapidly changing situations.”

Reviewer 2 Report

This is a timely paper; with constant increases in climate change-related disasters and their impact on communities as well as the rapidly evolving COVID pandemic and focus on precision health, people have increased interest in the factors that impact their health outcomes, with no additional training to understand all of the potential input. 

There are a few general concerns with this paper. 

While the older, white demographic is representative of Paradise, CA it may impact the generalizability of the study. Some of the sample communication materials were clear and repeated key messaging around chemicals that were/were not found (Figure 2) however were quite wordy and might benefit from being simplified, the same for the fact sheets (Supplemental figure 1). Although there were a few general rating comments about the communication materials, this was a big part of the project and if more information on how people responded to and understood the different components of the information provided, such as the depth of information and complexity of written materials, etc would be helpful to the reader.

Can the authors comment on the rather high number of participants that did not use email, and was not using email a general thing for those participants, unique to how they chose to interact with the study or the result of disaster-related disruptions?

The authors used multiple resource-heavy means by which to engage research subjects and convey results. While this is important to gather information on the best ways for other researchers to connect with their populations, more commentary on how this study experienced the different aspects of the 5 goals of effective results return (line 339) and the 12 'tips' (line 341) would be helpful for others. As well, delineating the additional factors needing to be considered (line 345) will help other researchers. 

More focus on how this team developed effective dissemination tools as well as materials would be helpful. As well, although this region was a relatively affluent, highly educated group, climate-related disaster will frequently impact socially and economically disadvantaged people. How can this material be adapted for communication with a broader audience?  If the team collected more detailed information about their communication tools from the participants, it could be used to inform commentary on this critical part of developing effective communication tools that has the potential to make the work more generalizable.

Author Response

We thank the reviewer for the helpful comments.

please see attachment.

Reviewer 3 Report

Some clarity may be needed concerning the "rolling basis" described in Line 160.  See comments in file.

Author Response

We thank the reviewer for the helpful comments.

COMMENT: Some clarity may be needed concerning the "rolling basis" described in Line 160. 

RESPONSE: We removed the term “rolling basis” because it was unclear and not necessary.  The results were sent out starting with high priority results (results with detections above or near an MCL).

Reviewer 4 Report

The manuscript by Von Behren et al. entitled "returning Individual Tap Water Testing Results to Research Study Participants after a Wildfire Disaster" explores how the quick collection of tap water to test contaminants in a population affected by wildfires and how the participants reacted to those results.

This manuscript summarizes a crucial research avenue, as frequently, participants are blind to the results of their tests. In this case, remediation involved the affected community. There are a few areas that may require some clarification:

Results:

Comment 1: 136 households were initially surveyed; however, only 90 (66%) completed the follow-up. Could you expand on the demographics of those who were excluded? Are there any characteristics of those households?

Comment 2: lines 323-326: Were you able to compare the results versus the water utility? How different were the results? Did you evaluate any trends in terms of remediation in your survey?

Comment 3: Are all the factsheets available? Those might be supplied as supplementary material or deposited somewhere as examples of your crisis package response.

Author Response

We thank the reviewer for the helpful comments.

COMMENT: Results line 136 households were initially surveyed; however, only 90 (66%) completed the follow-up. Could you expand on the demographics of those who were excluded? Are there any characteristics of those households?

RESPONSE:  We added this text at the end of section 3.2 to describe the non-responders: “The average age of the residents in the households that did not complete the follow-up survey was 48 years, which was similar to the responding households average age of 53 years. The participation rate was higher for residents served by the Paradise Irrigation District compared to the Del Oro Water Company (71% and 46% respectively), likely due to the heightened level of concern in residents served by PID.”

COMMENT: lines 323-326: Were you able to compare the results versus the water utility? How different were the results? Did you evaluate any trends in terms of remediation in your survey?

RESPONSE: Because of the relatively small number of participants from the Del Oro Water Company, we choose not to present detailed findings by water district.  We have added some additional information on the difference by water company to the text at the end of section 3.2 as follows:  “Most of the participants who completed the follow-up survey (N=77, 86%) received their water from the Paradise Irrigation District (PID), the largest provider in the area.  13 responding participants were served by the Del Oro Water Company (DOWC).   The responses were generally similar between residents in the two districts, except that more DOWC participants than PID participants reported that the research study information was inconsistent with what they were hearing from their drinking water utility and state agencies (23% vs. 9% respectively), which may have been due to the fact that the DOWC did not issue the same “do not drink/do not use” advisory that PID did.  Another difference we noticed between the participants in the two districts was that only one (8%) DOWC residents reported taking any action after receiving their tap water sampling results, compared to 31 PID presidents (40%).”

Comment regarding factsheets:  We could included one as supplemental material if the journal has space allowed.  Also, they can be made available through the NIEHS web site.